# Multi-Modal Deep Learning for Assessing Surgeon Technical Skill

**DOI:** 10.3390/s22197328

**Published:** 2022-09-27

**Authors:** Kevin Kasa, David Burns, Mitchell G. Goldenberg, Omar Selim, Cari Whyne, Michael Hardisty

**Affiliations:** 1Orthopaedic Biomechanics Lab, Holland Bone and Joint Program, Sunnybrook Research Institute, Toronto, ON M4N 3M5, Canada; 2Institute of Biomedical Engineering, University of Toronto, Toronto, ON M5S 1A1, Canada; 3Division of Orthopaedic Surgery, Department of Surgery, University of Toronto, Toronto, ON M5S 1A1, Canada; 4Division of Urology, Department of Surgery, University of Toronto, Toronto, ON M5S 1A1, Canada; 5Department of Surgery, Royal Victoria Regional Health Center, Barrie, ON L4M 6M2, Canada

**Keywords:** deep learning, surgical skills assessment, machine learning, computer vision, surgical education, biomedical engineering, multi-modal, human activity recognition

## Abstract

This paper introduces a new dataset of a surgical knot-tying task, and a multi-modal deep learning model that achieves comparable performance to expert human raters on this skill assessment task. Seventy-two surgical trainees and faculty were recruited for the knot-tying task, and were recorded using video, kinematic, and image data. Three expert human raters conducted the skills assessment using the Objective Structured Assessment of Technical Skill (OSATS) Global Rating Scale (GRS). We also designed and developed three deep learning models: a ResNet-based image model, a ResNet-LSTM kinematic model, and a multi-modal model leveraging the image and time-series kinematic data. All three models demonstrate performance comparable to the expert human raters on most GRS domains. The multi-modal model demonstrates the best overall performance, as measured using the mean squared error (MSE) and intraclass correlation coefficient (ICC). This work is significant since it demonstrates that multi-modal deep learning has the potential to replicate human raters on a challenging human-performed knot-tying task. The study demonstrates an algorithm with state-of-the-art performance in surgical skill assessment. As objective assessment of technical skill continues to be a growing, but resource-heavy, element of surgical education, this study is an important step towards automated surgical skill assessment, ultimately leading to reduced burden on training faculty and institutes.

## 1. Introduction

There has been a gradual evolution in surgical education towards objective assessment of competence as a requirement for trainee advancement and an increased reliance on simulation-based training [1]. This paradigm responds to mounting pressures to shorten the surgical trainee workweek, and improve operating room efficiency and safety at teaching institutions. However, competency-based medical education (CBME) can increase the burden on supervising surgical faculty and increase program reliance on the objectivity and validity of their CBME assessments [2].

Machine learning techniques, along with increased data-collection abilities across a variety of settings may offer the ability to tackle these challenges by automating some surgical skills assessments, potentially improving their objectivity and reducing the burden of CBME on training faculty and institutes. Deep learning in particular is well suited for tackling technical skills assessment due to its robustness to noise and flexibility to learn an optimal feature set representative of task performance from large, unstructured, and multi-modal data sources. Further, new innovations allow for the collection of large-scale multi-modal data in previously unwelcoming environments, such as the operating room [3].

However, existing work on surgical skills assessment has yet to fully exploit deep learning networks and large-scale data availability to automate skills assessment. Instead, previous research relies on classical machine learning algorithms [4], only classify high-level categories of performance [5], and rely on small datasets [6].

In this study, we investigate a unique multi-modal model to automate surgical skills assessment across multiple categories and evaluate its performance on a novel dataset. Specifically, our main contributions are as follows:Development of a multi-modal deep learning model that combines data from both images of the final surgical product and kinematic data of the procedure. We demonstrate that this model can assess surgical performance with comparable performance to the expert human raters on several assessment domains. This is significant since existing approaches are limited in scope and predominately focus on predicting solely high-level categories.Ablation studies comparing the image-based, kinematic-based, and combined multi-modal networks. We show that the multi-modal network demonstrates the best overall performance.A new dataset of seventy-two surgical trainees and surgeons collected during a University of Toronto Department of Surgery Prep Camp and Orthopaedics Bootcamp. This consists of image, video, and kinematic data of the simulated surgical task, as well as skills assessment evaluations performed by three expert raters. This large dataset will present new and challenging opportunities for data-driven approaches to surgical skills assessment and gesture recognition tasks. (The dataset can be downloaded here: https://osf.io/rg35w/ ).

In the following section we provide a brief synopsis of previous works related to surgical skills assessment and activity recognition. In Section 2 we describe the details of our data-collection, processing, and deep learning model development. We present the experimental results in Section 3, with a discussion of the results, comparisons with existing studies, and motivations behind the methodologies presented in Section 4. Finally, we summarize our main findings and discuss the broad impact of this work in Section 5.

### Related Work

Successful CBME is dependent on domain-specific assessment and feedback for trainees, as is currently provided by faculty members. Previous research in automating surgical skills assessment has shown promising results in effectively assessing global performance. For example, several recent studies [4,7,8,9] use machine learning techniques to classify surgical performance into “novice” or “expert” categories from kinematic time-series data. Other studies employ standard assessment frameworks, such as the Objective Structured Assessment of Technical Skills (OSATS) [10], to assess skill on various domains. However, many of these studies only classify performance in each domain into high-level categories (beginner, intermediate, advanced) [5,11,12]. Some studies do predict OSATS scores in a regression framework [11,13], however, the score prediction is only a small part of their work, and limited performance metrics are presented.

Instead of directly quantifying surgical performance, previous studies also focus on capturing proxies indicative of surgical performance, such as detecting surgical instruments [14], tracking instruments [15], or identifying events such as incisions [5]. Our work directly predicts the OSATS scores across five domains in a continuous regression framework. This is advantageous as it provides specific fine-grained assessment akin to that performed by a real faculty member, and eliminates ambiguities caused by broader discrete categories. Further, we present numerous performance metrics to understand the model’s performance, including direct comparisons with three expert human raters.

Machine learning algorithms have been applied to surgical skills assessment by previous studies [4,7,9,12,16]. Classical machine learning, combining engineered features with learned classifiers, as well as deep learning models have shown promising results for both skills assessment works [17], as well as human activity recognition tasks (HAR) [18,19,20,21,22,23]. More recent work has focused on deep learning networks because of their ability to better exploit rich data sources (e.g., images, videos, motion tracking), which has led to improvements in performance. The deep learning models applied generally use fully convolutional or convolutional-recurrent networks; leveraging one-dimensional convolutional layers as feature extractors and recurrent layers to capture temporal dependencies. This investigation expands upon the convolutional-recurrent networks [18,19] by applying a much deeper ResNet-18 based architecture, combined with a multi-modal approach. To our knowledge, no other works have reported leveraging deeper ResNet-LSTM based models to analyze kinematic data for surgical skills assessment tasks. We discuss this approach in more detail in Section 2 and Section 3.

Further, no existing studies use multi-modality approaches in surgical skills assessment. Multiple data sources (i.e., images of the final product, kinematic data of the procedure) can capture different information necessary for good performance across multiple domains of surgical skills assessment. Similarly to some image-based approaches [24], we employ a late-fusion approach. Some HAR studies investigate concatenating extracted features from different gestures for classical machine learning algorithms, and report that which features were extracted was more important than the fusion technique [23]. Unlike the studies in [23,24], we investigate fusing features extracted from disparate modalities (kinematic time-series + images) and not a single modality (images), and fuse learned features extracted from the raw data by the neural networks, instead of fusing hand-crafted features.

Previous investigations applying machine learning to surgical skills assessment have relied on small custom datasets, or the open-source JIGSAWS dataset [6]. The JIGSAWS dataset consists of video and kinematic data captured using a DaVinci Robotic system [25] from eight subjects (four beginner, two intermediate, two expert). These small datasets have presented a large limitation for data-driven methods such as machine learning. Many previous studies focus solely on data acquired using robotic systems or virtual simulators [5,11,17], and not on human-performed surgical tasks. In contrast, the dataset presented in this work larger and encompasses greater participant skill levels, containing 360 total samples from 72 participants across ten surgical divisions, with experience levels ranging from first year residents to staff surgeons. This challenging real-world dataset will enable new opportunities for research into automated surgical skills assessment. The dataset is described further in Section 2.

## 2. Materials and Methods

This project sought to develop and validate deep learning models for automated surgical skill assessment, specifically for the assessment of technical skill for a simulated knot-tying task. To facilitate this, 72 participants performed a knot-tying task, which were subsequently rated by human experts. Video and kinematic data of the task was recorded, as well as a photograph of the final product. In this study, the anonymized video recording was used for assessment by the human raters; the machine learning models used only image and kinematic data.

### 2.1. Surgical Task

Seventy-two surgical trainees and surgeons were recruited for participation in this study during the 2018 University of Toronto Department of Surgery Prep Camp and Orthopaedics Bootcamp suturing modules. Participants performed a simulated vessel ligature task using one-handed knot-tying with 0-silk ties on polypropylene tubing. Each participant performed the task five times consecutively, with each performance as a separate task. No feedback was provided to participants between executions of the task. The overall goal of this task is to determine if the trainees can correctly tie off, or occlude, the simulated blood vessel using the silk suture.

### 2.2. Data Collection

The vessel ligature tasks were recorded using three modalities, which are visualized in Figure 1:High resolution digital photograph of the final productAnonymized video recording of the operative field3D kinematic motion tracking of the hands using a Leap Sensor

### 2.3. Task Ratings

Three blinded independent raters conducted the technical skills assessment from the recorded video and photograph of the final product. The raters were senior surgical residents (PGY4 and above) with expertise in the assessed skill. Performance at the simulated surgical task was assessed by each rater using the Objective Structured Assessment of Technical Skill (OSATS) Global Rating Scale (GRS) [10] on the following four domains:1.Respect for Tissue2.Time and Motion3.Quality of Final Product4.Overall Performance

Each domain was scored on a 5-point scale (1–5). All raters were oriented to the OSATS GRS and domain specific anchors using example performances and suggested ratings. An example of the rating scale used by the human raters can be seen in Table 1.

It was also important to ensure that the dataset was collected from a diverse and representative set of participants, including diversity in aspects such as surgical division, and prior experience level. The plurality of participants were from the division of orthopaedics, with participants from nine other surgical divisions included. Most participants were Post-Graduate Year 1 (PGY1) trainees, with experience levels ranging up to Fellows and Staff surgeons. A summary of the experience level and surgical division of the participants can be seen in Figure 2.

The sequence of tasks was randomized so that the raters were not consecutively exposed to tasks performed by the same individual. Further, the randomization was seeded separately for each rater, providing each rater with a different random order of tasks to assess. Forty random samples were also selected to be rated a second time by each rater for test-retest reliability assessment.

### 2.4. Data Pre-Processing

The three-dimensional position data of each joint in the phalanges from both hands was extracted from the Leap Motion Sensor’s kinematic data capture. This 120 channel timeseries data was used as input into the deep learning models. The kinematic models require a fixed-length input, and the trials were not uniform in length. The Seglearn library [26] was used to truncate or zero-pad each data sample to a length of 4223 samples, which represents the 90th percentile of the sample lengths. This means that most samples were padded instead of truncated, so that as much information as possible was preserved. With a sampling rate of 110 Hz, this 4223-timestamp sequence is approximately 36 s long.

The Python implementation of OpenCV was used to pre-process the image data. The images were first temporarily masked to a binary image, isolating the black suture from the background. A dilating operation was applied to this image to enlarge the knot center. The OpenCV blob detector was then used to detect the suture knot, and a 512 × 512 bounding box was drawn around the center. The cropped image was then unmasked back to full RGB color. The kinematic and image data were also normalized between [0,1]. This is a standard deep learning procedure to speed computation time and avoid local minima in model optimization.

### 2.5. Data Augmentation

Although our dataset is not small relative to other relevant datasets, deep learning almost always benefits from larger quantities of data. Thus, the entire dataset was randomly oversampled to increase the number of training examples. Additionally, the trials with ratings that were greater or less than one standard deviation from the mean were further oversampled by a factor of three. By more evenly balancing the score distribution, the network can better learn to predict these minority classes.

However, increasing the size of the dataset without introducing any variation may lead to degraded performance, as the network may rely on memorizing specific features of the training data and fail to generalize to unseen data. Data augmentation may be used to alter the input instances, thus artificially increasing the variety of training data and the network’s ability to generalize. To minimize the model overfitting to the training data, the oversampled data was also augmented prior to input into the networks. The images were augmented with random 90-degree rotations and reflections about the x- or y-axis, largely to help mirror the varying knot orientation in the real data. The kinematic data was augmented based on recommendations in previous literature [27]: random rotations, reflections, and injection of Gaussian noise.

### 2.6. Machine Learning Models

We developed and analyzed three deep learning models. The first uses the RGB image data of the simulated vessel and ligature as input and the Quality rating as output. The second model uses the hand kinematic data as input and predicted the three other domains (Respect for Tissue, Time and Motion, and Overall Performance). The final is a composite model containing both RGB and kinematic modalities and output all four GRS rating domains. The video data was not used by the model.

The models were trained in a supervised regression learning framework, with the mean scores of the three expert raters as the ground truth. We trained the models to minimize a mean-squared error loss, however the number of output targets varied between the models since some predicted only one OSATS domain and others multiple.
(1)$$L=\frac{1}{N}\sum_{i=1}^{N}\sum_{j=1}^{K}\left(y_{i}^{k}-\hat{y_{j}^{k}}\right)$$

Here, *N* is the number of samples in the training batch, and *K* is the number of output targets. For example, the image-only model has a K=1 since it predicts only the Quality score, whereas the multi-modal has K=4 since all four domains are predicted.

Deep residual models (ResNets) are particularly powerful in training deeper neural networks with increased capacity to learn and model complicated relationships, achieving state-of-the-art performance on many image recognition tasks [28]. These improvements largely stem from the use of “skip connections”, or residual blocks, between layers which allow for deeper networks without suffering from vanishing gradient problems. This ability to effectively train very deep networks is the major advantage of the ResNet architecture. Although ResNet’s are often employed in image related tasks, they can also be implemented using one-dimensional convolutions for time-series data.

The image model is depicted in the bottom branch of Figure 3, and consists of a ResNet-50 backbone with pre-trained weights from the ImageNet dataset. Prior to input, the images were resampled to 1024 × 1024, further cropped 30% tighter, and normalized based on the ImageNet metrics. Following best-practises, the pre-trained networks were initially frozen for the first 200 epochs, and only the final dense layer was trained. This is to avoid the large gradient magnitudes from the new randomly initialized dense layer destroying the pre-trained weights [29]. Subsequently, the learning rate was reduced and the top layers of the ResNet model were fine-tuned for another 200 epochs. This freezing/fine-tuning method was followed for all subsequent pre-trained models and experiments.

Previous works demonstrate that convolutional-recurrent neural networks can been used to successfully perform human activity recognition from kinematic data [18,19]. In our work, the network was tasked with scoring surgical skill across multiple domains from a relatively high-dimensional dataset (120 channels). To ensure the network had the capacity to perform these tasks, a one-dimensional ResNet-18 model was used as a feature extractor on the kinematic data. The extracted features were then inputted into two bi-directional LSTM layers to model the temporal nature of the data. Finally, three dense layers were used to score the ‘Overall Performance’, ‘Respect for Tissue’, and ‘Time and Motion’ from the learned features. This model was trained for 200 epochs, and the architecture can be seen in the top branch of Figure 3.

The previous two models are combined so that all four GRS domains can be scored. The time series and image networks are trained concurrently, and the extracted feature sets are concatenated. These are then inputted into fully-connected layers to perform the final task scoring for each domain, as seen in Figure 3. The 2D ResNet network also leveraged pre-trained ImageNet weights and followed a fine-tuning scheme similar to that described above, where the ResNet layers were initially frozen for 50 epochs and used solely as a feature extractor, followed by fine-tuning the top layers of the ResNet for another 50 epochs.

The dataset was randomly split into 80%/10%/10% training/validation/testing sets. This means there were 58 participants (and 290 trials) in the training set, and 7 participants (35 trials) in the validation and testing sets. Further, the training epochs were tuned heuristically; we trained either until we saw substantial overfitting, or our computing resources were exhausted. Table 2 summarizes the hyper-parameters of the final multi-modal model.

### 2.7. Statistical Analysis

The collected dataset was analyzed to ensure its reliability and validity prior to being used for training and evaluating the deep learning models. The analysis of the expert human raters also serve as a baseline for understanding the model’s best achievable performance. The Intraclass Correlation Coefficient (ICC) and Standard Error of Measure (SEM) were used to analyze the human and AI ratings for agreement and consistency. To assess the interrater reliability on the entire collected dataset, the ICC (2,3), ICC(2,1), and SEM scores were calculated for each of the GRS domains [30]. The ICC (2,3) model is selected since our raters are chosen as representative of a larger population, and the mean of the three raters is used as the ground-truth. The ICC (2,1) was also used to assess the human raters on their test-retest consistency, using the randomly repeated trials that were rated twice. Our hypothesis was that the human raters show moderate to good agreement on the GRS domains and good consistency in their ratings.

In addition to measuring the average human rater reliability on the entire dataset, we also looked at the ICC score of the raters on the held-out testing subset of the data. Since the AI models were evaluated on this test set, finding the human rater’s reliability on this subset alone can allow for a more direct comparison with the network performance.

The experience levels of the participants and their ratings were also investigated to help establish construct validity. A one-way ANOVA was performed between the beginner (PGY1 & PGY2, n = 48), intermediate (PGY3, PGY4, & PGY5, n = 18), and expert (Staff & Fellow, n = 6) level participants. A Tukey–Kramer post hoc test was then done to determine which groups were different from each other. These tests were all done using the participants performance on the “Overall Performance” GRS domain.

Several tests were done to evaluate the model’s performance. The point difference between the model’s predictions and the human ratings with the ground truth was evaluated using the mean squared error (MSE). The goodness of fit of the model was evaluated using $R^{2}$. Finally, the agreement amongst each (human or AI) rater and the ground truth was determined using the ICC (2,1) score. This means that the ICC between the AI ratings and the ground truth was determined, as well as the ICC between each human rater and the ground truth. This allows us to consider how our model performs as a single generalized rater [30] in terms of its agreement with the ground-truth data, as well as compare the AI agreement with that exhibited by the humans. Our hypothesis was that the AI would demonstrate comparable point errors (MSE) and agreement (ICC) with the ground truth data as the human raters.

Although previous research seeking to directly predict GRS scores is sparse, existing studies report performance using the mean Spearman Correlation Coefficient ρ across the predicted vs. true GRS scores [13]. For consistency in the reported metrics, we also evaluate the Spearman Coefficient on the multi-modal model.

Finally, some studies that directly predict the GRS domain scores report their performance in terms of accuracy [11]. For a comparable metric, we also find the accuracy of our multi-modal model. Since our predictions are continuous and accuracy deals with discrete data, we first round the ground-truth and model predictions; for example, a score of 2.7 will get rounded to 3.0, which is necessary to compute the accuracy metric. Our model is designed to predict continuous scores so this is not a perfect metric, but serves to gain a general comparison with previous studies.

## 3. Results

### 3.1. Dataset Analysis

The human raters showed ICC scores corresponding to moderate agreement on the four GRS domains, when measured on the entire collected dataset, as summarized in Table 3.

There was some variance in the test-retest performance of the human raters, with ICC scores ranging from 0.49 to 0.88, and SEM ranging from 0.37 to 0.58. Overall, Rater 1 demonstrated better consistency amongst their ratings than Rater 2 or 3. Although some raters performed better than others, overall, they all showed moderate to good consistency, and the results are summarized in Table 4.

On the held-out test set, the human raters showed good to excellent agreement, as seen in Table 5. Greater agreement was seen on this smaller subset of the overall data likely because there are fewer samples for the human raters to disagree on.

The one-way ANOVA returned a *p*-value of 0.0038, suggesting there was a significant performance difference amongst the surgeon experience groups. The Tukey analysis resulted in a significant difference between the Beginner (n = 48, mean = 2.31) and Intermediate (n = 36, mean = 2.79) groups (*p* = 0.003), and no significance between the Expert group (n = 6, mean = 2.50) and either of the two groups. The lack of significance in the Expert group may be due to the relatively small sample size compared to the other two. The results of the ANOVA are depicted in Figure 4.

### 3.2. Deep Learning Model Performance

The kinematic, image, and multi-modal models were all trained and evaluated independently of each other on the same reserved testing set. The model performance was evaluated by how well it can predict the mean OSATS GRS ratings provided by the raters, as well as the intrarater reliability between the model predictions and the expert raters.

Table 6 highlights the performance relative to the ground-truth. For a direct comparison with the human performance, the same metrics are presented for each individual rater’s score compared to their mean scores, for the test-set trials. These metrics serve as an understanding for how close the model predictions are to the dataset’s ground truth. The model’s predictions do appear close to the ground-truth, with lower point errors than two of three human raters, and with the multi-modal model exhibiting the lowest point error overall.

The error between the ground-truth and the model predictions, as well as human ratings, is also seen in Figure 5. The improvements of the multi-modal model were particularly noted on the Overall Performance domain.

Table 7 summarizes the agreement between the AI model and the ground truth scores (i.e., mean of the human ratings). For comparison, we also considered the ICC scores between each individual rater and their mean score. The AI model demonstrated ICC scores ranging from 0.3 to 0.90, with the human raters ranging from 0.60 to 0.92. The multi-modal model demonstrated better agreement based on the ICC and SEM than the kinematic or image-only models on all domains except for Respect for Tissue. The multi-modal model also demonstrated better agreement with the ground truth than 2 of the 3 human raters on the Overall Performance and Quality of Final Product domains, however its performance was poorer on the remaining two domains.

The Spearman Correlation Coefficient, ρ, of our multi-modal model is reported in Table 8. This represents the correlation between the model’s predictions and the ground truth.

The discretized scores are used to evaluate the model’s accuracy, and are summarized in Table 9. As mentioned, accuracy is not a perfect metric for our continuous data predictions, however it is indicative of the difference between the predictions and ground-truth on the datasets.

Overall, the multi-modal model demonstrated comparable results to the humans on most of the GRS domains. The AI had a lower point error on the ground truth scores than the human raters on three of the four GRS domains, as exhibited by the lower MSE. The ICC metrics suggest that in general, the human raters were in better agreement with the ground-truth scores. The multi-modal model demonstrated the best performance, with higher ICC on some domains (e.g., Quality of Final Product) than two of the three raters.

## 4. Discussion

This paper presented a new dataset consisting of multi-modal recordings (image, video, & kinematic) of a simulated surgical knot-tying task, with skill assessment conducted by expert human raters based on the OSATS GRS framework. A thorough statistical analysis was conducted to ensure the validity of the dataset. Three deep-learning models were trained and evaluated on this dataset: a ResNet-50 image model, a unique “ResLSTM” kinematic model, and a combined multi-modal model.

All three models were able to successfully perform the skills assessment, with the multi-modal model performing the best overall. In comparison to previous studies conducted on the JIGSAWS dataset [6], which contains video and kinematic data from eight surgeons performing three surgical actions (knot tying, needle passing, and suturing) using the DaVinci Robotic System [25], our multi-modal model achieves better performance. For comparison, previous literature report a mean Spearman Correlation of ρ=0.65 on the knot-tying task in the JIGSAWS dataset [13], as seen in Table 8. This means that on average, our multi-modal model demonstrates better correlation between its predictions and the ground-truth on our dataset, than reported on similar datasets in previous literature. Further, Khalid et al. [11] present a study that directly predicts the GRS scores in a regression fashion, using the video data of the JIGSAWS dataset. As seen in Table 9, they report a mean accuracy of 0.32 for Time and Motion, 0.51 for Quality of Final Product, and 0.41 for Overall Performance.

This is particularly encouraging as assessing surgical skill from human performed knot-tying is seemingly more challenging than evaluating a robotically operated dataset. This result means that our model can be used in a wider range of environments and facilities, where robotic surgery systems are not available for surgical trainees or faculty. Further, while some studies attempt to indirectly compute performance metrics for surgical skill [14,15], our model directly predicts performance on the GRS domains and provides the most pertinent assessment of surgical skill to trainees.

The AI performance was comparable to the human rater on three out of the four GRS domains. Further experiments are required to determine why the model consistently struggles on the Respect for Tissue domain. A possible explanation is that since the Leap Sensor is only tracking the subjects’ hands, important information on the handling of the “tissue” (or polypropylene tubing) is not captured using this modality. Respect for Tissue was better assessed on video which was available to raters but not used by the model. Future analysis will investigate leveraging the video modality within the multimodal model to improve performance on this domain.

The image-only model was trained solely on the Quality of Final Product domain, since it is not likely that the images alone contain enough relevant information for this model to perform well on the other categories (e.g., Time and Motion). Smaller models were investigated for this task, such as 5- and 7-layer convolutional neural networks, however these all exhibited poor performance in the rating task and were abandoned. This suggests that the ResNet’s increased capacity to extract important and meaningful features from the image data is important in assessing surgical skill. We also explored using a pre-trained MobileNet as the imaging backbone, however found the performance to be poorer than the ResNet-50. The ResNet-50 presents a good balance between performance (better ImageNet performance than VGG [28]), and reasonable computing requirements. Future studies may investigate the use of alternate backbone networks, including models such as Vision Transformers [31].

Similarly, shallow recurrent neural networks exhibited poor performance on the kinematic data and were also discarded. Learning to score various categories of surgical skill is a complex task and these models likely did not have the capacity to extract the necessary features from the kinematic data. This justifies the development of a deeper, more powerful “ResLSTM” model; the one-dimensional ResNet-18 backbone and bi-directional LSTM layers exhibited far better performance on our dataset than shallower networks. This outperforms a LSTM-only network for two likely reasons: (1) the ResNet extracts meaningful features from the raw sensor data, and (2) the convolution operators reduce the length of the time-series sequences, which are easier for the LSTM layers to learn than longer sequences.

Leveraging transfer learning was also important to increasing the image model’s performance. Training a ResNet-50 model without weights pre-trained on ImageNet leads to an RMSE of 0.523 (0.274) for the quality of final product score, compared to the RMSE score of 0.392 (0.146) exhibited with pre-training. Although the ImageNet dataset does not contain examples of surgical sutures, the low-level features learned on the large-scale generic dataset are helpful starting points when transitioning to a domain-specific task. Our results further suggest the need for even larger datasets that can be used for pre-training the kinematic portion of the model. The image only model performed better than the kinematic model, likely in part due to the availability of ImageNet pre-trained weights for the image feature extractor.

Combining both the kinematic and image modalities allows for a single model to rate all four surgical skill assessment categories. Further, training a single model on both modalities led to an increase in performance across all the categories, except for Respect for Tissue. It is unclear why this model sees a degradation in performance in this category compared to the kinematic-only model; further experiments are required to discern this. Notably, the Overall Performance category saw a large increase in MSE and $R^{2}$ scores. Training on both kinematic and image data allows for the combined model to learn a more optimal feature set that is better representative of the task performances.

This study is limited in that the AI was trained and evaluated on data collected from a single training center. It remains to be studied how the model performance is affected by increased participant diversity, e.g., trainees from different institutes or countries. Future studies can investigate how the model generalizes to new participants. Further, while the OSATS was used in this study to evaluate the knot tying performance, improved assessment tools, such as a modified OSATS score which incorporates additional domains [32], may be more suitable in future studies as more complex tasks are considered in more physiologically challenging environments.

## 5. Conclusions

This study demonstrated a multi-modal deep learning model for surgical skill assessment with performance comparable to expert raters. This investigation highlights the importance of multi-modal data sources (image, kinematic, video) in surgical skill assessment. Automation of surgical skill assessment has the potential to transform surgical education, making training more effective, equitable, and efficient; trainees can receive quicker and more frequent feedback, while surgical faculty will have less of a burden to evaluate, allowing for greater focus on educational and clinical tasks. Further, with the addition of data collection systems to the operating room, skill assessment technology has the potential to lead to greater surgeon skill and improved patient outcomes.

## Figures and Tables

**Figure 1 sensors-22-07328-f001:**
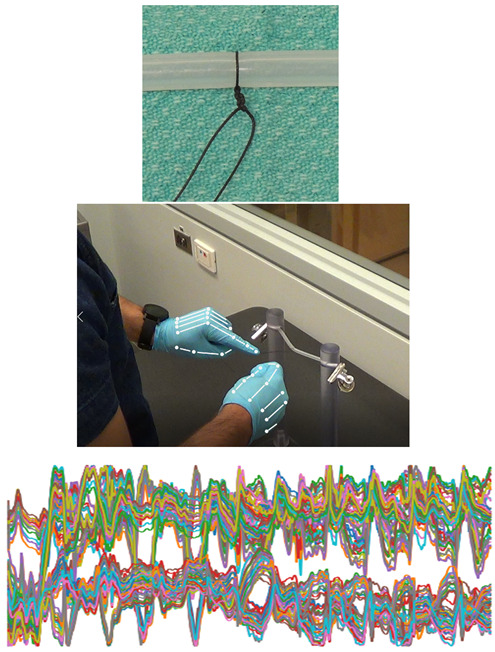
The trials were recorded using three modalities. The top is an image of the final product, the middle is a screen capture of the video data with a visualization of the joints tracked by the Leap sensor. The bottom is an example of the kinematic time series data, representing the temporal 3-dimensional movement of the hand joints during the knot tying task.

**Figure 2 sensors-22-07328-f002:**
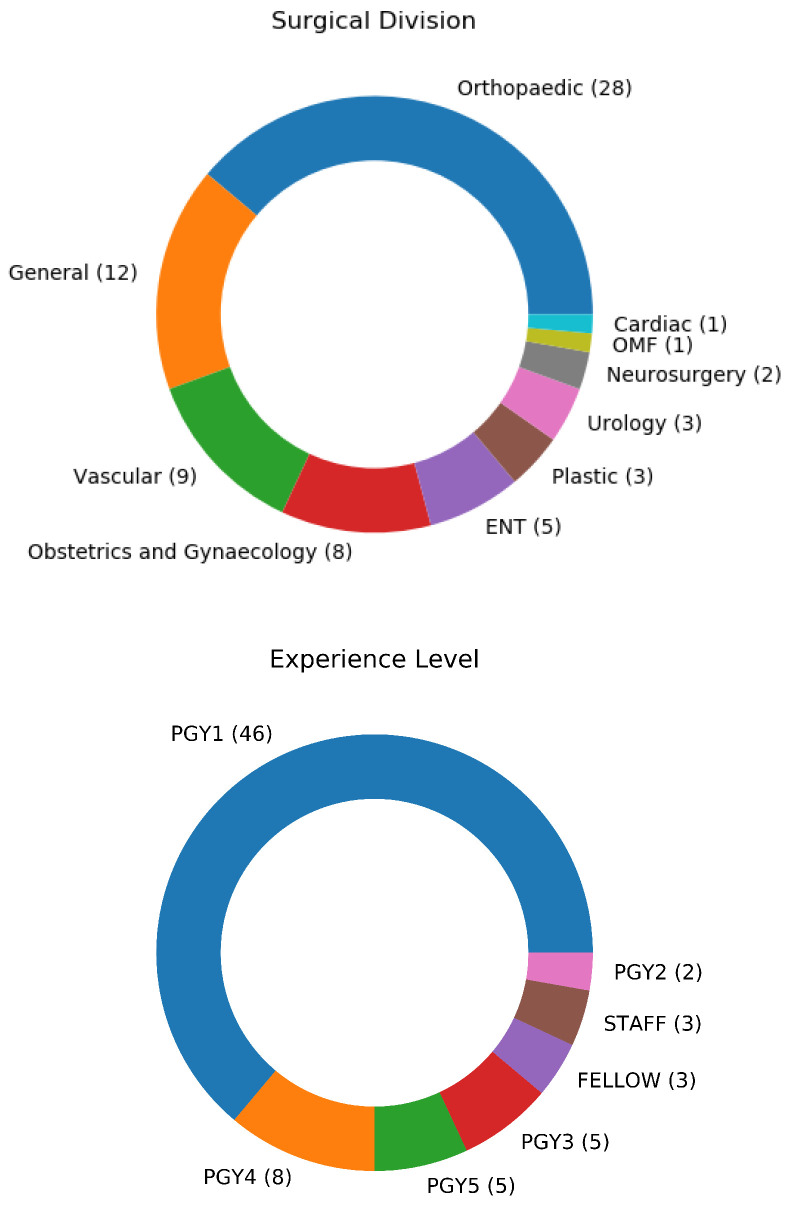
Participants came from 10 surgical divisions, with experiences ranging from PGY1 to Fellow.

**Figure 3 sensors-22-07328-f003:**
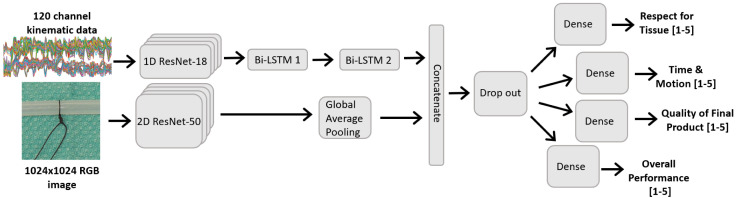
Images were analyzed using a ResNet-based network, and the kinematic data was analyzed using a 1D ResNet-18 as a ‘feature extractor’, followed by 2 bidirectional LSTM layers. The combined multi-modal network is concurrently trained on both the image and kinematic data as input, and predicts all four GRS domains.

**Figure 4 sensors-22-07328-f004:**
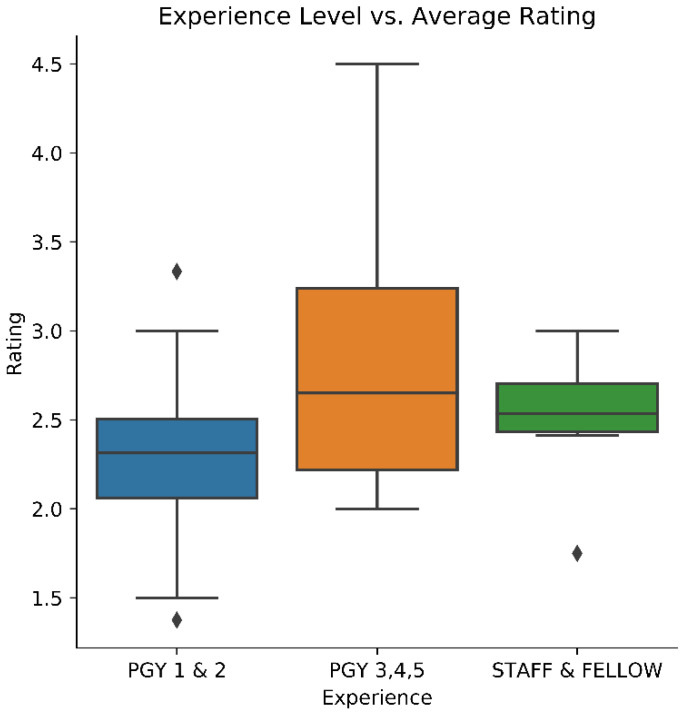
Participant experience and rating on the `Overall Performance’ domain. A significant difference was found between the Beginner and Intermediate groups.

**Figure 5 sensors-22-07328-f005:**
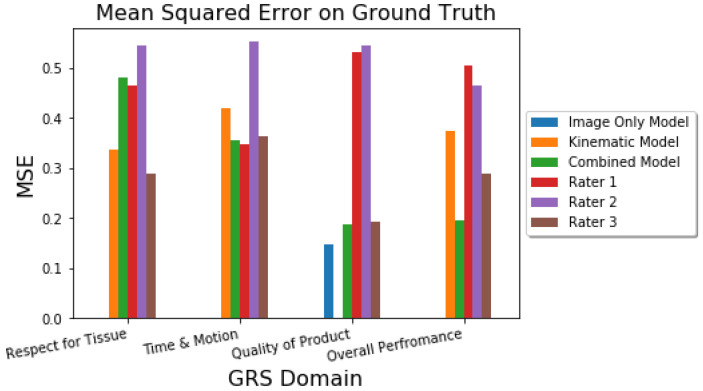
Graphical comparison of the MSE on the GRS Domains—lower MSE is better.

**Table 1 sensors-22-07328-t001:** Rating scale used when evaluating surgical skill on the GRS Domains.

Domain	Rating Scale
Respect for Tissue	1—Very poor: Frequent or excessive pulling or sawing of tissue
	3—Competent: Careful handling of tissue with occasional sawing or pulling
	5—Clearly superior: Consistent atraumatic handling of tissue
Time and Motion	1—Very poor: Many unnecessary movements
	3—Competent: Efficient time/motion but some unnecessary moves
	5—Clearly superior: Clear economy of movement and maximum efficiency
Quality of Final Product	1—Very poor
	3—Competent
	5—Clearly superior
Overall Performance	1—Very poor
	3—Competent
	5—Clearly superior

**Table 2 sensors-22-07328-t002:** Summary of the hyper-parameters used to train the multi-modal network. Hyper-parameters were tuned heuristically.

Hyperparameter	Value
Learning rate	$1\cdot 10^{-4}$
Optimizer	Adam ($\beta_{1}=0.9,\beta_{2}=0.999$)
Batch size	16
Dropout	0.50
Epochs (frozen backbone)	50
Epochs (fine-tuning backbone)	50
Loss function	Mean Squared Error
Image dimensions	(1024, 1024)
Timeseries length	4223 timestamps

**Table 3 sensors-22-07328-t003:** The expert human raters demonstrate moderate to good agreement on their evaluations when as measured using the mean. The AI model was trained & evaluated on the mean value of the ratings.

GRS Domain	ICC (2,3)	SEM (2,3)	ICC (2,1)	SEM (2,1)
Respect for Tissue	0.71	0.45	0.47	0.62
Time and Motion	0.70	0.47	0.44	0.64
Quality of Final Product	0.83	0.40	0.63	0.61
Overall Performance	0.73	0.39	0.47	0.55

**Table 4 sensors-22-07328-t004:** Test-retest performance of the human raters on the forty repeated trials. Although the raters performance varies, they all show moderate to good consistency.

GRS Domains	Rater 1		Rater 2		Rater 3	
	ICC	SEM	ICC	SEM	ICC	SEM
Respect for Tissue	0.84	0.43	0.49	0.55	0.55	0.54
Time and Motion	0.83	0.46	0.57	0.58	0.62	0.48
Quality of Final Product	0.88	0.40	0.79	0.47	0.69	0.43
Overall Performance	0.85	0.37	0.60	0.49	0.58	0.48

**Table 5 sensors-22-07328-t005:** Human raters show good to excellent agreement on the held-out test set. Determining agreement on the same test set the AI model is evaluated on can help provide a better baseline for expected performance.

GRS Domain	ICC (2,3)	SEM (2,3)	ICC (2,1)	SEM (2,1)
Respect for Tissue	0.78	0.44	0.54	0.63
Time and Motion	0.81	0.41	0.58	0.61
Quality of Final Product	0.93	0.30	0.82	0.49
Overall Performance	0.86	0.30	0.68	0.30

**Table 6 sensors-22-07328-t006:** Performance metrics, including mean squared Error (MSE), of the AI predictions and human ratings, compared to the ground truth (mean of human scores).

Model	Metric	Respect for Tissue	Time and Motion	Quality of Final Product	Overall Performance
Image Model	MSE	-	-	**0.146**	-
	RMSE	-	-	0.392	-
	MAE	-	-	0.293	-
	R2	-	-	0.778	-
Kinematic Model	MSE	**0.336**	0.420	-	0.373
	RMSE	0.579	0.648	-	0.610
	MAE	0.523	0.456	-	0.431
	R2	**0.337**	0.244	-	0.453
Multi-modal Model	MSE	0.480	**0.356**	0.186	**0.194**
	RMSE	0.693	0.597	0.431	0.440
	MAE	0.545	0.459	0.331	0.315
	R2	0.136	**0.476**	**0.838**	**0.618**
Rater 1	MSE	0.464	**0.348**	0.531	0.505
	RMSE	0.681	0.590	0.729	0.710
	MAE	0.528	0.474	0.449	0.407
Rater 2	MSE	0.546	0.553	0.545	0.466
	RMSE	0.739	0.744	0.738	0.683
	MAE	0.586	0.483	0.425	0.436
Rater 3	MSE	**0.288**	0.363	0.193	0.290
	RMSE	0.537	0.602	0.439	0.539
	MAE	0.409	0.426	0.291	0.336

**Table 7 sensors-22-07328-t007:** Intraclass Correlation Coefficient (ICC) and Standard Error of Measurement (SEM) scores between the ground truth and the AI models & human raters.

Model	Metric	Respect for Tissue	Time and Motion	Quality of Final Product	Overall Performance
Image Model	ICC(2,1)	-	-	0.888	-
	SEM(2,1)	-	-	**0.257**	-
Kinematic Model	ICC(2,1)	**0.477**	* **0.621** *	-	0.534
	SEM(2,1)	**0.464**	0.441	-	0.416
Multi-modal Model	ICC(2,1)	0.301	0.591	**0.904**	* **0.746** *
	SEM(2,1)	0.499	**0.428**	0.309	**0.305**
Rater 1	ICC(2,1)	0.717	0.779	0.823	0.616
	SEM(2,1)	0.476	0.414	0.512	0.502
Rater 2	ICC(2,1)	0.606	0.627	0.758	0.508
	SEM(2,1)	0.516	0.524	0.521	0.689
Rater 3	ICC(2,1)	**0.797**	**0.797**	**0.924**	**0.789**
	SEM(2,1)	**0.377**	**0.423**	**0.308**	0.379

**Table 8 sensors-22-07328-t008:** Spearman Correlation Coefficient between the multi-modal AI predictions and the ground truth. Best performing model on the JIGSAWS dataset included as reference [13].

GRS Domain	ρ	
	Multi-Modal Model (Ours)	FCN [13]
Respect for Tissue	0.18	-
Time and Motion	0.73	-
Quality of Final Product	0.95	-
Overall Performance	0.82	-
Mean	**0.67**	0.65

**Table 9 sensors-22-07328-t009:** Accuracy of the multi-modal model, determined by first rounding the continuous ground-truth and predicted scores. Best performing model on the JIGSAWS dataset included as reference [11].

GRS Domain	Accuracy	
	Multi-Modal Model (Ours)	Embedding Analysis [11]
Time and Motion	**0.54**	0.32
Quality of Final Product	**0.76**	0.51
Overall Performance	**0.76**	0.41

## Data Availability

The dataset and code will be made publicly available and shared once posted online.

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
