# Peer review of "Multi-Modal Deep Learning for Assessing Surgeon Technical Skill"

_sensors, 2022, doi:10.3390/s22197328_

Round 1

Reviewer 1 Report

The authors present a new knot tying data set, expert ratings of the knot tying, and machine learning models that estimate the knot tying quality. The size and diversity of the data set is excellent, including three different levels of experience and many different surgical divisions.

The abstract is well written and provides an excellent summary of the paper. The description of the materials and methods is very good. The data preprocessing is also well documented. The authors describe how they handle different duration signals and how they resample the image data. The architecture of the machine learning models is well described. A little more detail on the number of neurons in each layer of the network would be helpful. The authors compare their combined model to the two component models and show that the combined model outperforms its components. The division of training and testing data is not provided. This is needed.

The results show good consistency between the combined model and the expert raters. The discussion section is very useful and compares the authors’ results to the JIGSAW data set.

Author Response

Question 1: The division of training and testing data is not provided. This is needed.

Response: We thank the reviewer for their comments & feedback. The data was randomly split into an 80%/10%/10% training/validation/testing division. This means there were 58 participants (and 290 trials) in the training split, and 7 participants (35 trials) in the validation and testing splits. Please refer to final paragraph of Section 2.6 in the updated manuscript. 

Reviewer 2 Report

The paper proposes a deep learning method for assessing surgeon technical skills. The paper is rather weak. The authors should motivate better their choices and make more clear technical details. I strongly suggest a major revision.

My comments are reported below. I will use the bold font for pointing at lines or sections. I will use the italic font for reporting verbatim sentences from the paper that raise me some concerns. I will enumerate every comment to address.

Comments

1. [...] Deep learning offers the ability to tackle these challenges by automating some surgical skills assessments, potentially improving their objectivity and reducing the burden of CBME on training faculty and institutes. [...]

Only deep learning does it? I think this is a too strong statement.

2. Introduction: What is the novelty of the paper behind proposing a new dataset?

3. Materials and methods/Introduction: Is the dataset available?

4. Introduction: What are the contributions of this paper to the research field?

5. Introduction/Related works: What are the gaps filled in the literature? What gaps have not been filled yet? A related work section is mandatory. Authors should discuss in detail all previous works in touch with this one, in terms of techniques used, research field, context and so on. I also suggest to look at this paper for multi-modal approaches and compare with: https://doi.org/10.1007/s00521-022-07454-4https://doi.org/10.1371/journal.pone.0245230 

6. Section 2.6: I'm not sure about the novelty of the multi-modal system used. Is there any algorithmic insight concerning the deep learning approach?

7. Section 2.6: Why the use of ResNet? Please motivate the choice. Furthermore I strongly suggest to compare the developed model with others in literature. For example, authors could use VGG19 and DenseNet instead of ResNet.

8. Section 2.6: Please show all the hyper-parameters of the nets.

9. Section 2.6: Please, motivate the choice of the epochs to train the nets.

10. Section 2.6: Did the authors perform validation phase? How the validation has been conducted? It is hard to grasp from the paper.

11. Section 3.2:  Authors must compare their method with others in literature over different aspects: energy, ram, time, MSE, R2...

12. Section 3.2: Why only using MSE and R2? Why not RMSE, NRMSE and MAE?

13. Conclusion: what is the expected impact of your solution? Please, provide a brief paragraph about it.

Furthermore, could be useful for readers not familiar with deep learning, a brief introduction to ResNets and the other neural networks used in this paper in the section materials and methods.

Author Response

We thank the reviewer for their comments and constructive feedback. Please see below our responses to the review comments:

Point 1: Only deep learning does it? I think this is a too strong statement.

Response: We have acknowledged that other approaches, deep learning as well as the ubiquity of data collection in a variety of settings has made these challenges more easily addressable.

Point 3: Introduction: What is the novelty of the paper behind proposing a new dataset?

Point 4: Introduction: What are the contributions of this paper to the research field?

Response: 
We have made the contributions clearer in the introduction by explicitly stating them, quoted below:

  1. Development of a multi-modal deep learning model that combines data from both images of the final surgical product and kinematic data of the procedure. We demonstrate that this model can assess surgical performance with comparable performance to the expert human raters on several assessment domains. This is significant since existing approaches are limited in scope and predominately focus on predicting solely high-level categories.
  2. Ablation studies comparing the image-based, kinematic-based, and combined multi-modal networks. We show that the multi-modal network demonstrates the best overall performance.
  3. A new dataset of seventy-two surgical trainees and surgeons collected during a University of Toronto Department of Surgery Prep Camp and Orthopaedics Bootcamp. This consists of image, video, and kinematic data of the simulated surgical task, as well as skills assessment evaluations performed by three expert raters. This large dataset will present new and challenging opportunities for data-driven approaches to surgical skills assessment and gesture recognition tasks.

Point 4: Materials and methods/Introduction: Is the dataset available?

Response: The dataset will be made publicly available. 

Point 5: a) Introduction: What are the gaps filled in the literature? What gaps have not been filled yet?

We have narrowed the introduction to specifying the problem of surgical skill assessment and the promise of this field with new data-driven tools and the ability to collect.  We have now provided a Related Work section discussed below.

b) Related Works: A related work section is mandatory. Authors should discuss in detail all previous works in touch with this one, in terms of techniques used, research field, context and so on. I also suggest to look at this paper for multi-modal approaches and compare with: https://doi.org/10.1007/s00521-022-07454-4, https://doi.org/10.1371/journal.pone.0245230

Response: 

We have added a related work section that more thoroughly discuses prior related work.  We further highlight how this investigation builds upon the previous work in this section.  We have made specific reference to the papers suggested by the reviewer; we discuss the similarities and differences between their approach and ours. For example:

Similarly to the image-based approaches of Seeland & Mader, we employ a late-fusion approach. However, unlike their study, we investigate fusing features extracted from disparate modalities (kinematic time-series + images), and not a single modality (images).

As mentioned in the introduction, most existing studies focused on automating surgical assessment rely on robotically operated data and not human performed tasks. Further, they assess performance in broad categories (beginner, intermediate, expert) and do not assess on the OSATS domains, as a real surgical faculty member would. Existing studies also present limited comparisons with human raters, and unlike our work, rely on single modalities (e.g., either only images or only kinematic data).

We thank the reviewer for the suggested papers.

Point 6: Section 2.6: I'm not sure about the novelty of the multi-modal system used. Is there any algorithmic insight concerning the deep learning approach?

Response: 

To our knowledge, no existing studies use multi-modality approaches in surgical skills assessment. Multiple data sources (i.e., images of the final product, kinematic data of the procedure) can capture different information necessary for good performance across multiple domains of assessment for surgical performance. Similarly to the image-based approach in the suggested paper (https://doi.org/10.1371/journal.pone.0245230), we employ a late fusion approach. However unlike that study, we investigate fusing features extracted from disparate modalities (kinematic time-series + images), and not a single modality (images).

As we demonstrate, the combined multi-modal network outperforms the single-modality networks, suggesting that the different modalities are in fact helpful in achieving good performance. We discuss this in the related works section of the updated manuscript. We have added to our discussion that the results further suggest the need for even larger datasets that can be used for pre-training the kinematic portion of the model. The image only model performed better than the kinematic model, this was likely due to the availability of ImageNet pretrained weights for the image feature extractor.

Point 7: Section 2.6: Why the use of ResNet? Please motivate the choice. Furthermore I strongly suggest to compare the developed model with others in literature. For example, authors could use VGG19 and DenseNet instead of ResNet.

Response: 

We have included other attempted strategies and the motivation for the ResNet in Section 4 – Discussion. Further, we briefly investigated using a pre-trained MobileNet as the image backbone. However, the ResNet-50 exhibited better performance, so we continued with the ResNet model. The ResNet-50 presents a good balance between performance (better ImageNet performance than VGG), and ease of implementation/computing resources. Nonetheless, future studies may certainly investigate the use of alternate backbone networks, including model such as ViT or ResNeXt. We have updated Section 4 to include this information.

We have included in the Discussion, shallow convolutional-recurrent networks (3-6 convolutional layers) were investigated for the kinematic data, however these exhibited poor performance. This motivation is further expanded in the Discussion; the larger 1D ResNet-18 model is able to extract more meaningful features from the kinematic data, compared to the shallower networks.

Point 8: Section 2.6: Please show all the hyper-parameters of the nets.

Response: We have included  Table 2 in section 2.6 of the updated manuscript for a hyper-parameter summary.

Point 9: Section 2.6: Please, motivate the choice of the epochs to train the nets.

Response: The number of epochs was tuned heuristically, training until we began to see substantial overfitting. Further, the pre-trained weights were initially frozen to prevent large gradient magnitudes from new layers destroying the pre-trained weights during backpropagation. Please refer to Section 2.6 in the updated manuscript.

Point 10: Section 2.6: Did the authors perform validation phase? How the validation has been conducted? It is hard to grasp from the paper.

Response: The data was randomly split into an 80%/10%/10% training/validation/testing division. This means there were 58 participants (and 290 trials) in the training split, and 7 participants (35 trials) in the validation and testing splits. This has been included in the final paragraph of Section 2.6 in the updated manuscript.

Point 11: Section 3.2:  Authors must compare their method with others in literature over different aspects: energy, ram, time, MSE, R2.

Response: Table 7 and Table 8 compare our results with other studies that use deep learning approaches for surgical assessment.

Point 12: Section 3.2: Why only using MSE and R2? Why not RMSE, NRMSE and MAE?

Response: Please refer to Table 5 in the updated manuscript for additional performance metrics, including RMSE and MAE.

Point 13: Conclusion: what is the expected impact of your solution? Please, provide a brief paragraph about it.

Response: 

We have revised the conclusion to the following:

This study demonstrated a multi-modal deep learning model for surgical skill assessment with performance comparable to expert raters. This investigation highlights the importance of multi-modal data sources (image, kinematic, video) in surgical skill assessment. Automation of surgical skill assessment has the potential to transform surgical education, making training more effective, equitable, and efficient; trainees can receive quicker and more frequent feedback, while surgical faculty will have less of a burden to evaluate, allowing for greater focus on educational and clinical tasks. Further, with the addition of data collection systems to the operating room, skill assessment technology has the potential to lead to greater surgeon skill and improved patient outcomes.

We have also updated Section 2.6 to include additional background information on ResNets.

Round 2

Reviewer 2 Report

I appreciate the effort of the authors in revising the paper and providing answers to all my comments. Great work!

I still have a couple of concerns:

1. I asked whether the dataset was available or not, since it is one of the major contributions of the paper as stated by the authors, but the response was "it will be made available", so I assume is now not available, hence how can it represent one of the major contributions? When will it be available?

2. The authors have compared their approach with those proposed by Seeland & Mader, but not on those proposed in https://doi.org/10.1007/s00521-022-07454-4 or other papers using similar techniques. I suggest to broadening such a comparison.

Author Response

  1. I asked whether the datasetwas available or not, since it is one of the major contributions of the paper as stated by the authors, but the response was "it will be made available", so I assume is now not available, hence how can it represent one of the major contributions? When will it be available?

We have updated the manuscript to include a footnote with the URL where the dataset can be found:

The dataset can be downloaded here: https://osf.io/rg35w/

  1. The authors have compared their approach with those proposed by Seeland & Mader, but not on those proposed in https://doi.org/10.1007/s00521-022-07454-4 or other papers using similar techniques. I suggest to broadening such a comparison.

We have updated the Related Works section to explicitly compare / discuss the paper by Guarino et. al:

Some HAR studies investigate concatenating extracted features from different gestures for classical machine learning algorithms, and report that which features were extracted features was more important than the fusion technique (Guarino et. al). Unlike the studies in (Seeland & Mader, Guarino et. al.), we investigate fusing features extracted from disparate modalities (kinematic time-series + images) and not a single modality (images), and fuse learned features extracted from the raw data by the neural networks, instead of fusing hand-crafted features.

We also mention the work by Guarino et. al. in the Related Works section when discussing other machine learning approaches in human activity recognition:

Classical machine learning, combining engineered features with learned classifiers, as well as deep learning models have shown promising results for both skills assessment works, as well as human activity recognition tasks (HAR) (Guarino et. al., …).